# Medical Interpreting Services for Refugees in Canada: Current State of Practice and Considerations in Promoting this Essential Human Right for All

**DOI:** 10.3390/ijerph21050588

**Published:** 2024-05-02

**Authors:** Akshaya Neil Arya, Ilene Hyman, Tim Holland, Carolyn Beukeboom, Catherine E. Tong, Rachel Talavlikar, Grace Eagan

**Affiliations:** 1Department of Family Medicine, McMaster University, Hamilton, ON L8S 3L8, Canada; 2School of Public Health Sciences, University of Waterloo, Waterloo, ON N2L 3G1, Canada; catherine.tong@uwaterloo.ca; 3Kitchener-Waterloo Centre for Family Medicine Refugee Health Clinic, Kitchener, ON N2G 1C5, Canada; cbeukebo@uwo.ca; 4Dalla Lana School of Public Health, University of Toronto, Toronto, ON M5T 3M7, Canada; i.hyman@utoronto.ca; 5Department of Bioethics, Dalhousie University, Halifax, NS B3H 4R2, Canada; timothy.holland@dal.ca; 6Newcomer Health Clinic, Halifax, NS B3L 4P1, Canada; 7Arthur Labatt Family School of Nursing, Western University, London, ON N6A 3K7, Canada; 8Lawrence S. Bloomberg Faculty of Nursing, University of Toronto, Toronto, ON M5T 1P8, Canada; 9Department of Family Medicine, University of Calgary, Calgary, AB T3H 0N9, Canada; rachel.talavlikar@albertahealthservices.ca; 10Mosaic Refugee Health Clinic, Calgary, AB T2A 5H5, Canada; 11Language Services & Digital Strategy, Access Alliance Multicultural Health & Community Services, Toronto, ON M5T 3A9, Canada; geagan@accessalliance.ca

**Keywords:** refugees, newcomers, medical interpretation/interpreting, language barriers, medical ethics, healthcare standards, human rights for all, Canada

## Abstract

Language barriers, specifically among refugees, pose significant challenges to delivering quality healthcare in Canada. While the COVID-19 pandemic accelerated the emergence and development of innovative alternatives such as telephone-based and video-conferencing medical interpreting services and AI tools, access remains uneven across Canada. This comprehensive analysis highlights the absence of a cohesive national strategy, reflected in diverse funding models employed across provinces and territories, with gaps and disparities in access to medical interpreting services. Advocating for medical interpreting, both as a moral imperative and a prudent investment, this article draws from human rights principles and ethical considerations, justified in national and international guidelines, charters, codes and regulations. Substantiated by a cost-benefit analysis, it emphasizes that medical interpreting enhances healthcare quality and preserves patient autonomy. Additionally, this article illuminates decision-making processes for utilizing interpreting services; recognizing the pivotal roles of clinicians, interpreters, patients and caregivers within the care circle; appreciating intersectional considerations such as gender, culture and age, underscoring the importance of a collaborative approach. Finally, it provides recommendations at provider, organizational and system levels to ensure equitable access to this right and to promote the health and well-being of refugees and other individuals facing language barriers within Canada’s healthcare system.

## 1. Introduction

The provision of medical interpreting services is essential to meet the needs of the thousands of refugees and other newcomers who arrive in Canada each year, many of whom experience language barriers. In 2017, according to Statistics Canada, 19.6% of Canadian permanent residents who arrived as immigrants spoke neither English nor French and this proportion rose to 49% among refugees [1]. Refugees often face the need for services to prevent and address immediate health challenges on arrival and in the immediate post-migration period [2]. With such limited proficiency in the host country’s language and without interpreting services, these people are put needlessly at risk [3]. As we describe below, such access nonetheless varies widely across Canada and is often denied to refugees (among other populations) seeking healthcare.

The aim of this paper is to make a solid case for the provision of medical interpreting services, particularly for refugees, from our perspectives as clinicians and service deliverers, ethicists and health equity-focused researchers, accompaniers and advocates. We first describe the state of medical interpreting services across Canada including inconsistencies and current gaps. Next, we argue that interpretation is a right and obligation, not a privilege, and that the provision of interpreting services makes good economic sense. Then, to assist decision makers in setting up interpreting services, we review current options available and key considerations in their adoption. Finally, we provide recommendations for the implementation of interpreting services at the individual, provider and system levels.

Note that we use the term “medical interpreting” throughout, as opposed to medical interpretation, to emphasize the active, dynamic and interpersonal nature of this work. 

## 2. The Current State of Interpretation across Canada: A Patchwork of Care

Twenty years ago, Sarah Bowen identified ‘newcomers’ (immigrants and refugees) as among four groups in Canada who face language barriers accessing healthcare when their primary language is not an official language; others being Indigenous community members, deaf persons and francophones or anglophones in minority language situations [4]. For the purposes of this paper, we are focusing on the heightened needs of refugees. In this work, Bowen noted that while access to necessary health services is a right of every Canadian—referring to the Canada Health Act of 1984—“access has generally been interpreted to mean the absence of explicit financial barriers to care.” [4].

The provision of medical interpreting services is inconsistent across Canada and there has been relatively little progress in reducing language-related barriers to care. Healthcare services are coordinated and delivered by the ten provinces and three territories and there is considerable variation in the ways in which these jurisdictions approach and fund medical interpreting. Appendix A summarizes the provision of such services in Canada’s ten provinces, where most of the Canadian population, and nearly all refugees, live. For this section, we have used our own experience and that of colleagues from our Canadian Refugee Health Network [5]. Further, we received more direct information from our colleagues including five from each of Ontario and Quebec, three from each of British Columbia, Alberta and Saskatchewan and one from each of Manitoba, New Brunswick, Nova Scotia and Newfoundland. While most were family doctors, we also had two psychologists, an academic, a provincial administrator, two practicing in public health, one psychiatrist and one pediatrician. Many provinces have delegated the coordination of care to regional authorities (referred to with terms, depending on the province, such as regional health authorities, local health integration networks, provincial health teams, etc.), with different authorities/regions opting for different types of medical interpreting and different funding models. 

In some provinces, such as Nova Scotia and Prince Edward Island (PEI) (and more recently in Manitoba where the Winnipeg-based service has been expanded to the entire province, and where trained casual employees are now working for Shared Health), the provincial government pays the cost of the telephone interpreting service for all physicians. Other provinces, such as Alberta, offer telephone services for centralized primary care services such as HIV clinics, Indigenous clinics and hospital-based ambulatory clinics. Newfoundland provides interpreting services for health authority clinics but expects fee-for-service physicians to cover these costs in community clinics. 

Ontario and Quebec provide no overall provincial coverage, instead relying on individual hospitals, health authorities and institutions to develop interpreting services independently, often from their own operating budgets. Local health authorities have assumed this responsibility in larger newcomer-receiving cities, such as Toronto and Ottawa; however, these may only cover a limited number of providers within their jurisdiction. 

Some regional and local health clinics in Ontario have independently raised funds for medical interpreting. For example, Community Health Centres in Kitchener, Hamilton, Toronto, Ottawa and Windsor; academic family health teams in Ottawa; francophone centres in a few cities and the Centre for Family Medicine (CFFM) Refugee Health Clinic in Kitchener and, for a time, more generally in Windsor (WE Speak) using (Remote Interpretation Ontario (R.I.O.) Network and, later, LanguageLine Solutions’ (LL) Interpreter on Wheels. This again highlights the precarity and inherent inequities in the provision of interpreting services. 

The predominantly francophone province of Quebec has a unique system that includes two “banques” of qualified interpreters (civil servants and freelance), which healthcare organizations (hospitals, community health centres and academic institutions) and providers can access for both in-person and telephone services. However, public funding for these ‘banques’ is not protected and healthcare organizations may use their allocated interpretation budgets for other needs (with some exceptions, in which the budgets are protected, for example, at specific refugee health clinics in larger cities such as Montréal, Quebec City and Sherbrooke) and in Montreal by the CIUSS (integrated health and social services centre) du Centre-Ouest de l‘Île de Montréal (covering CLSCs (local community service centres) Parc Extension and Côte des Neiges which serve large immigrant and refugee populations). 

Hospitals across the country grapple with inconsistent medical interpreting policies and funding models, with variability provincially and even within cities. Toronto hospitals generally have access to R.I.O. Network. However, in Montreal, McGill hospitals may use R.I.O. Network and are piloting VOYCE, another service based in the US which may have easier access, especially for video, but does not provide interpretation into French, while on the French side, they may use the ‘banques’, which do not provide interpretation into English. Quebec’s pediatric hospitals have access to the aforementioned ‘banques’ of interpreters, (with Montreal Children’s even having its own in-house SCIS bank of interpreters), whereas adult hospitals and community health services are left without paid professional interpreters, relying on a mixture of family, friends and ad hoc interpretation with multilingual healthcare workers. Pediatric hospitals such as IWK in Nova Scotia and the Children’s Hospital of Eastern Ontario (CHEO) in Ottawa have budgets for telephone and sometimes video and in-person services but adult hospitals in the same cities may not. 

In some cases, where institutions have failed to provide interpreting services, community and private sector organizations have intervened. For example, medical students at McGill University, in Quebec, developed a Phone Tree in 2016 to support care in teaching sites, from primary care to hospitals. Immigrant, ethno-cultural and religious organizations and non-governmental organizations (NGOs) across the country have stepped in to fill the many gaps in this ‘system’. Individual businesses have also stepped in. For example, an imaging company in Calgary, Alberta, provides interpretation and a small pharmacy in Southern Ontario chose to provide services in Arabic because of a large clientele base. However, funding and personnel for such ventures are precarious [6,7]. Finally, with rare exceptions of special allocated funding, allied health professionals generally are not covered outside of hospitals and occasionally others in the governmental sector. 

## 3. Making the Case for Medical Interpreting

“All human beings are born free and equal in dignity and rights. They are endowed with reason and conscience and should act towards one another in a spirit of brotherhood.”—Article 1—UN Universal Declaration of Human Rights [8].

### 3.1. Why Interpretation Is an Essential Right

We assert that the right to interpretation stems from four fundamental human rights: (1) Right to healthcare; (2) Right not to face discrimination; (3) Right to informed consent/refusal and (4) Right not to be harmed. Support for these rights is drawn from United Nations (UN)/World Health Organization (WHO) documents, professional codes and legal frameworks. Consistent with the ethical arguments made, this section also includes empirical evidence to show that medical interpreting improves health, quality of life and operational outcomes. 

#### 3.1.1. The Right to Healthcare

The WHO recognizes healthcare, conceptualized in terms of four core components—Availability, Accessibility, Acceptability and Quality [9]—as a human right [10]. When a language discordance exists between a provider and patient and interpreting services are not available, none of these core components are satisfied.

The relationship of language discordance to healthcare outcomes is well documented in the literature [11,12]. For example, a Canadian study exploring language discordance between providers and elderly patients in hospital settings found that patients with language barriers experienced a higher risk of adverse effects, an increase in in-hospital deaths and longer hospital stays [13]. A study in two Toronto hospitals found that patients with both acute and chronic diseases who faced language barriers were more likely to return to the emergency room (ER) within 30 days of a previous visit, as well as be readmitted [14]. Chronic conditions (congestive heart failure and chronic obstructive pulmonary disease) require comprehensive discharge instructions along with medication management, thus, if not properly explained, may lead to poorer outcomes [14]. The authors noted that the use of appropriate interpretation would also improve patient health, safety and experience, cost efficiency and provider satisfaction. 

Karliner and colleagues’ [15] systematic review of 21 studies provided further evidence that the provision of professional medical interpreting services improved care in four key areas: improved clinical outcomes, improved healthcare utilization, improved communication (including comprehension and the reduction in medical errors) and improved patient satisfaction. Flores’ [16] systematic review of 36 studies similarly found that professional interpreting supported better clinical outcomes, perceived quality of care and patient satisfaction. Flores’ work also emphasized the critical importance of interpretation for specific conditions and concerns, such as mental health. Specific to refugees, a scoping review of 84 peer-reviewed studies suggests that using professional interpreting services for refugees also improves clinical outcomes and satisfaction with care [17], although that review was exclusively focused on mental health providers. Taken together, these reviews demonstrate the diverse ways in which professional medical interpreting improves quality of care. A comprehensive chart describing the benefits of addressing and not addressing language barriers related to better health outcomes, improved patient experience, improved staff experience and cost savings can be found in Hyman’s 2021 review [18]. For refugee patients, particularly those navigating trauma and resettlement, the benefits of interpretation cannot be overstated.

Despite the WHO’s designation of healthcare as a right, and clearly improved health outcomes associated with interpretation in that setting, some critics would argue that since resources are always limited, it is unreasonable to oblige society to enforce such ‘positive’ rights (i.e., *provision of goods and services* as in the right to clean water, food or even healthcare) [19]. The only actionable rights such proponents would contend are ‘negative’ rights, to be *protected* from the actions of another, such as the right not to be killed, tortured or have your goods stolen from you [19]. In the case of healthcare, it would mean that in order to not suffer unjustified discrimination, once a service was provided it should be of no lower quality than for others in the same jurisdiction. Should we grant this distinction, there are still grounds for a right to interpretation. 

#### 3.1.2. The Right Not to Face Unjustified Discrimination

Even if the right to healthcare were debatable, it is clear that governments in most industrialized nations have accepted a responsibility to provide healthcare, as evidenced by the considerable resources such nations dedicate to the provision of healthcare within their borders [20]. Thus, we assert that the failure to ensure that patients with language barriers are able to access equitable healthcare services amounts to discrimination. 

The right not to be discriminated against unjustifiably is instantiated in a large range of documents across many jurisdictions. Looking most broadly, at the United Nations’ Declaration Universal of Human Rights, Article 21.2 clearly states that “*Everyone has the right of equal access to public service in his country*.” [8] Healthcare services provided by a country to its residents are, by definition, a public service. Further, Article 2 of that same document states, “*Everyone is entitled to all the rights and freedoms set forth in this Declaration, without distinction of any kind, such as race, colour, sex, **language**, religion, political or other opinion, national or social origin, property, birth or other status*” [8]. Language discordance does not appear as an essential feature that justifies discrimination to access healthcare. Further, though not listed explicitly in Section 15 of the *Canadian Charter of Rights and Freedoms* [21], language is closely tied to national and ethnic origin, and discrimination on that basis is widely recognized as morally reprehensible and protection from such discrimination IS widely accepted as a fundamental right, while Section 14, for legal proceedings, explicitly states that “A party or witness in any proceedings who does not understand or speak the language in which the proceedings are conducted or who is deaf has the right to the assistance of an interpreter”. 

When we consider access to care for analogous populations being a human right, provincial legislation may be relevant. The Ontario Human Rights Code (OHRC) [22] promotes equal rights and opportunities and prohibits discrimination on various grounds, including disability. The OHRC and the Accessibility for Ontarians with Disabilities Act (AODA) [23] each recognize the importance of effective communication for individuals with disabilities. Each requires organizations, including healthcare and service providers, to provide accessible services, equal treatment and accommodation for such individuals. In the case of communication for persons who are deaf, hard of hearing or who have speech difficulties, this may include sign language, oral or other forms of interpretation. The OHRC does not specifically identify “language” as a prohibited ground of discrimination but may consider claims under a number of related grounds, such as ancestry, ethnic origin or country of origin. The AODA similarly does not explicitly ensure the provision of interpreting services but encourages organizations to be proactive in identifying and removing barriers to communication including providing interpreting services such as sign language interpreters, captioning or other means of facilitating communication based on individual needs. It seems only a small step to require such services for refugees who experience language barriers.

Focusing more specifically on healthcare professionals, the Canadian Medical Association’s 2018 revision of the Code of Ethics and Professionalism has added specific reference to language in Professional Responsibility 1: “*Accept the patient without discrimination (such as on the basis of age, disability, gender identity or expression, genetic characteristics, **language**, marital and family status, medical condition, national or ethnic origin, political affiliation, race, religion, sex, sexual orientation, or socioeconomic status).*” [24] This revised Code has been adopted by the regulatory colleges in British Columbia, Alberta, Manitoba, Nova Scotia, Newfoundland and Labrador [25,26,27,28,29]. 

The Canadian Pediatric Society recently issued a statement declaring the following: 1. Trained face-to-face interpreters and video or telephone interpreting services should be available in hospitals and other healthcare settings where patients and physicians are not proficient in the same language. 2. Children and youth should not be used as interpreters in healthcare settings. 3. Interpreting services should be part of hospital accreditation standards and organizations that represent health professionals and agencies responsible for accreditation should work with the interpretation community to develop and implement a national standard for interpreting services. Finally, establishing free 24 h interpreting services should be a priority for all provinces and territories [30].

While not specifically making interpreting services “mandatory” in healthcare settings, various healthcare professional bodies in Ontario have attempted to incorporate the provision of language and interpreting services within their documents and policies related to code of conduct, standards of practice, ethics, consent to treatment, the charter of rights and responsibilities, therapeutic patient relationships, privacy and confidentiality, patient-centred care and patient safety [31]. A comprehensive review related to interpreting services within the various professional colleges in Ontario is available from the KW4 Ontario Health Team *An environmental scan of interpretation done in Kitchener, Ontario,* which researched various health professionals’ regulatory requirements related to language interpretation [31].

#### 3.1.3. The Right to Autonomy—Informed Consent/Refusal

The right to autonomy is encapsulated in the right of patients to give free and informed consent for treatment or to refuse any treatment, often simply termed “informed consent” [32]. The necessary elements of an informed consent process include reliable information on diagnoses, disclosure of benefits and risks of treatment options and plans and a safe space to ask questions and have questions answered. Many key guiding documents and global partnerships highlight the ethical values of participation, inclusion, transparency and accountability. It is important to note that the informed consent and refusal process is the cornerstone for all four of these values [33].

An example was related to one of us by a patient experiencing language barriers who presented to the emergency room with a miscarriage and accepted a medication without knowing its name, purpose or effects. Thus, when provider–patient language discordance is present, without an appropriate interpreting service, informed consent or refusal, for example, to surgery cannot be attained, and the right to autonomy, as enshrined in the right to informed consent/refusal, would be denied.

The importance of interpreting services is explicitly stated in many guiding documents. For example, the College of Physicians and Surgeons of British Columbia (CPSBC) has published guidelines for consent to treatment when there are equity considerations. This document does not specifically mandate the use of interpreting services but provides a very strong indication of the merits of these services. It also points out that interpreting services are legally required for persons who are deaf or hard of hearing, similar to what was outlined above in the Ontario context [34]. While interpretation has been identified as an accessibility issue for some groups (e.g., those who are deaf or hard of hearing), it seems to us that the conceptualization of interpretation as an accessibility issue has not been appropriately extended to those experiencing language barriers with “limited English proficiency” (LEP). 

#### 3.1.4. The Right Not to Be Harmed

All humans have an inherent right not to be harmed. In Canada, this is instantiated as the right to security of the person within the *Charter of Rights and Freedoms* [21] and within the *Canadian Medical Association’s Code of Ethics and Professionalism* as “Take all reasonable steps to prevent or minimize harm to the patient.” [24]. It is well documented that the failure to provide interpreting services in healthcare is often associated with adverse health events and unjustified harm [18,35,36].

For example, patients experiencing language barriers may undergo unwanted procedures due to violations of the informed consent process [37,38,39,40]. The failure to use interpreting services may result in inadequate history taking and an increase in diagnostic tests with their associated costs, non-adherence to treatment regimens, frequent emergency room re-visits, re-hospitalization and medical errors causing poor health outcomes and even death. Unfortunately, patients with precarious status and language barriers are often disempowered and consequently are less likely to make malpractice claims. An analysis of malpractice claims in four US states found that 2.5% of all claims were related to inadequate use of interpreting services [41]. The inherent power imbalance in the clinician–patient relationship is accentuated by the absence of an interpreter and may be perceived by some as a micro-aggression.

### 3.2. Making the Case: Cost/Benefit Case for Interpretation Especially for Refugees

Research has shown that medical interpreting offers a myriad of benefits including better patient, physician and institutional outcomes. Intuitively, patients who receive healthcare services in a language they understand experience better quality care and health outcomes [42,43,44,45,46]. For example, if a patient cannot describe their symptoms, their healthcare provider will not be able to gather the necessary information to make an accurate diagnosis. If a patient cannot understand their pharmacist’s instructions for their medications, they are far less likely to adhere to the intended regimen. If a healthcare provider cannot communicate effectively with their patient, they will be unable to develop a trusting therapeutic relationship, nor can patients freely express their healthcare needs with their medical team. 

Further, the literature identified several factors specific to refugees that must be considered when addressing the provision of medical interpreting services to this group. For example, a significantly higher proportion of refugees to Canada, compared to immigrants, report physical, emotional or dental problems [47], and a decline in health status post-migration [48]. Refugees often experience difficulties in expressing physical and mental health symptoms and require greater resources and time for cultural orientation, education on basic aspects of healthcare services and prevention and mental health screening [17]. Low health literacy among refugee patients is associated with challenges in adherence to medical recommendations [49]. Mangrio’s 2017 scoping review article recommended that refugee patients be provided with sufficient information about the healthcare system, in both oral and written formats, and have their rights to healthcare guaranteed in resettlement countries [50].

However, few economic analyses have been conducted to enumerate direct (financial) and indirect costs, for example, poor quality of care leading to adverse events and readmissions. 

Some health service decision makers maintain that cost-benefit is an important consideration for the implementation of interpreting services [40,51,52,53,54]. However, some institutions do not keep track of the costs of the interpreting services they provide, and it is challenging to identify, document and quantify all possible costs and consequences of not providing interpreting services, some of which may be long-term. 

A seminal study by Hampers et al. found that the overall mean charge for diagnostic tests was significantly higher for patients with a language barrier compared to those without (USD 145 vs. USD 104) [55]. Jacobs et al. compared the cost of utilization of primary care and emergency department services before and after the introduction of professional interpreting services and concluded that providing interpreting services is a financially viable method for enhancing the delivery of healthcare to patients experiencing language barriers [56]. An older study by Jean-Baptiste et al. [57] found that LEP patients stayed 6% (approximately 0.5 days) longer overall than EP patients at three Toronto hospitals from 1993 to 1999. Similar findings have been reached in other studies [58,59]. For example, Karliner et al. demonstrated that a systems intervention aimed at increasing access to telephone interpreters decreased readmission rates and estimated hospital expenditures [60]. The intervention consisted of providing in-patients with dual-handset telephones with a direct connection to interpreting services at each hospital bedside. During the eight-month intervention, the estimated net savings equalled USD 1,291,233 for an estimated monthly healthcare expenditure saving of USD 161,404.

Findings from a US study that showed that the provision of professional interpreting services at both admission and discharge reduced a patient’s length of hospital stay by 34% or 0.75 days [61] and were recently applied to the Ontario context. Using this figure to estimate savings per patient, researchers determined this would result in a savings of USD 860 per patient, which is significantly higher than the total cost of an interpreter [18,62].

## 4. Interpretation Options and Decision Making

### 4.1. How to Determine When Interpretation May Be Required? 

When do *we* need an interpreter? Before answering this question, it is important to consider who the “we” in the question is. It is not only the patient who does not speak English or French but also the provider who does not speak the patient/client’s language. This allows us to explore the question as part of achieving effective communication where ideas are exchanged via messages that are sent, received and understood by the interacting parties. Secondary to the question is then asking who is qualified to interpret, which will be explored in the next section.

Researchers have examined the need for medical interpreting from the perspective of clinicians, interpreters, patients and caregivers [63,64,65]. While the patient’s perspective is paramount, the perspectives of all members of the circle of care are also important and each brings different concerns and considerations. 

Family members often play a significant role in medical interpreting, whether they wish to or not, though guidelines mandate the use of professional interpreting services, not *ad hoc*/family interpreters [66]. It is essential that clinicians include caregivers in the decision-making process about the need for, and preferred modes of interpretation, and the extent to which caregivers wish to be involved [67]. While caregivers may be present to facilitate communication, they also often serve as cultural brokers and ‘cultural guardians’ for the patient [64]. Interpreters also bring important insights to clinical scenarios. In some instances, they support the continuity of care and system navigation, as the interpreter may accompany the patient through the healthcare system (e.g., from primary care to a specialist appointment, then back to primary care) [63]. 

The Institute of Medicine recommends using a screening process to determine the preferred language. “Screening for preferred language is a fundamental component in any measurement strategy related to quality improvement in language services.” [68]. First, ask “What language do you feel most comfortable using when speaking with your healthcare provider?” and subsequently ask the patient to self-assess their ability to speak and understand English. If the patient answers anything other than “very well” for health-related discussions, interpretive services should be utilized for the patient’s preferred language [69].

Decision support tools are available to determine when the use of interpretive services is required. One such tool is a rubric of services provided by the agency together with a rating of the potential risk of compromised communication [70]. For example, in a hospital setting, bedside care is on one end and obtaining informed consent for treatment is on the other end. The agency then determines where on that spectrum front-line staff must engage an interpreter to support a discussion in a language-incongruent situation [70].

Frontline staff may also be provided with a list of settings/contexts in which interpretive services must be utilized in the context of language barriers. This list would include high-risk clinical scenarios, such as emergency care, informed consent, surgical care, medication reconciliation and discharge [70].

### 4.2. Types of Interpretation

Despite clear evidence that the provision of interpreting services improves patient care and clinical outcomes, there continues to be confusion about ‘how’ interpretation should be provided and in ‘what format’ (remote vs. in-person and by whom) [15]. Thus, whether as front-line staff, healthcare professionals, administrators or regulatory bodies, we must consider how to choose among the types of interpreting available.

#### 4.2.1. Training and Role

The *Canadian National Standard Guide for Community Interpreting Services* by the Healthcare Interpretation Network [71] defines a spectrum of interpreters ranging from the completely untrained “*Ad Hoc*”, “lay” or “chance” family, friend, staff or volunteer; “*Accredited*”, one who has been awarded a certain recognition or accreditation having passed the screening criteria of a particular organization; *Professional Interpreter*, a fluently bilingual individual with appropriate training and experience who is able to interpret with consistency and accuracy and who adheres to the *Standards of Practice and Ethical Principles* and *Certified Interpreter*, a professional interpreter who is certified as competent by a professional organization through rigorous testing based on appropriate and consistent criteria. Interpreters who have had limited training or have taken a screening test administered by an employing legal, health, interpreter or referral agency are not considered certified [72].

Factors that policy makers and providers should consider when implementing interpreting services include determining when an interpreter is needed, understanding the training and role of interpreters, selecting the most appropriate mode of interpreting and being mindful of intersectional factors, such as gender and age. These considerations are outlined more fully in the sections below. 

We generally recommend working with professional rather than ad hoc interpreters. Although the use of family members (especially children) as interpreters is a common practice, it is problematic and generally discouraged because of the emotional burden that interpreting for a loved one might cause, biases introduced, the lack of confidentiality and concerns about the quality of the interpretation, particularly when using unfamiliar medical terms [73,74]. Obligations for the protection of private health information may not be as defined when ad hoc interpretation is used, the clarification of which is possible using certified interpreters or accredited interpreters who will have signed contracts. This also contributes to the trust that the patient and provider can place that the information being shared is accurate and reliable, particularly when health decisions depend on it [75]. In an emergency room setting, when ad hoc interpreters were used the number of errors was actually higher compared to no interpreter at all (and was predictably decreased when professional interpreters were used) [76]. In our experience and in the literature, providers tend to engage interpreters who are the quickest and most accessible [77,78,79]. Family members, peer health navigators or other untrained individuals within the facility or a bilingual healthcare provider may seem a simple solution to interpreting needs. While these sources may offer comfort to patients, they provide the lowest or most variable quality of interpreting and professionalism when compared to trained and/or regulated interpreters, where access and funding then become the issues [72].

#### 4.2.2. Developing and Implementing Standards for Interpreting Services

There are two perspectives to consider when developing and implementing standards: interpreting service provision and interpreting service utilization. Provision includes interpreters, interpreting service providers and interpreter accreditation/certification frameworks; utilization explores end-users (health/public service providers and consumers with limited English proficiency) and the purchasers/procurers of interpreting services. 

While national Canadian standards have existed since 2007 [71] and are the basis for interpreting service provider (ISP) certification and standardized assessment testing is available in most spoken languages (our review in August 2023 at the time of submission found that though major bodies offered such services (ILSAT—Interpreter Language and Skills Assessment Tool (63 languages), CILISAT—Community Interpreter Language and Interpreting Skills Assessment Tool (70 languages), MAG—Ministry of the Attorney General court interpretation service (24 languages) and CTTIC—Canadian Translators, Terminologists and Interpreters Council (11 languages)) community interpreting remains an unregulated industry with no requirement for ISP certification. Spoken-language community/healthcare interpreter training programs vary widely—from single-day orientation sessions to 180 h college certificate programs. Though interpreters have an ethical obligation to ongoing education and professional development, this is only enforced or monitored at the discretion of hiring ISPs or by certifying or accrediting bodies. It is at the discretion of the ISP to determine the minimum qualifications required and who is qualified to interpret for specific assignments and modalities. Working conditions for interpreters (financial compensation, minimum booking duration, short-notice cancellation protection, travel/mileage compensation, debrief support and professional development support) vary widely, sometimes due to differences in local realities and almost always at the discretion of the hiring ISP. Certifying and accrediting bodies are growing in membership and provide a mechanism for arm’s length credentialing, i.e., separate from the training institutions and hiring ISPs. Based on the directories published on the provincial certifying/accrediting/regulatory websites, it is estimated that there are fewer than 450 interpreters in Canada certified or accredited as community and/or health/medical interpreters. 

* Note that this was a mere headcount based on a review of provincial bodies such as ATIO (The Association of Translators and Interpreters of Ontario), ATIA (The Association of Translators and Interpreters of Alberta), ATIS (The Association of Translators and Interpreters of Saskatchewan), ATIM (Association of Translators, Terminologists and Interpreters of Manitoba), ATINS (Association of Translators And Interpreters of Nova Scotia), CTTIC (Canadian Translators, Terminologists and Interpreters Council), CTINB (Corporation of Translators, Terminologists and Interpreters of New Brunswick), OCCI (Ontario Council on Community Interpreting), OTTIAQ (Quebec Order of Translators, Terminologists and Certified Interpreters) and STIBC (Society of Translators and Interpreters of British Columbia) by author G.E. in August 2023.

### 4.3. Modes of Interpretation: In Person, Telephone and Video Format

Patients require access to interpreting services at various points during their medical journey, for example, at the time of booking or confirming a medical appointment, upon arrival for check-in, when being “roomed” or reviewed by nursing, during the actual appointment and then afterwards for support with health navigation whether going to a lab or pharmacy. Even if Google Translate were available, using an online booking system, even for English speakers, requires a certain literacy level. Patients may use or require different types of interpreters during a single visit, where formal services might be used by the clinician with ad hoc support from family or friends when booking investigations or follow-up appointments. 

In the past, in-person methods of interpreting were most commonly provided; however, with the advent of technology, the use of telephone, video, web-based and even simultaneous interpreting formats are increasingly employed [63,80], with varying levels of effectiveness and acceptability. Some studies suggest that patients are grateful to obtain care, regardless of the type of interpreting used, and do not report strong preferences for in-person or remote interpreting. Many studies confirm that receiving care in a congruent language is strongly preferred and simultaneous translation is associated with feeling respected [81,82,83,84]. Heath et al.’s recent systematic review of different types of interpreting in healthcare settings (professional, ad hoc, relational and any or no interpreter) [85] found that in-person professional interpreting results in the highest levels of patient satisfaction. 

While these findings confirm that ‘any’ interpreter is better than ‘no’ interpreter, confidentiality issues may arise in the case of the provision of interpreting services in smaller communities where community members all know each other [86]. Some research conducted among providers and interpreters suggests that in-person and video-based formats are preferred for more sensitive conversations including family conferences, or in situations where cultural or body language and nuance were beneficial to providing care including visits relating to mental vs. physical health [87]. As recently as 2020, the ISO (International Organization for Standardization) published Standard 21,998 [88], which is meant to assist in the choice of healthcare interpreting services.

Remote communication services play a vital role in ensuring the continuity of healthcare services during the COVID-19 epidemic. Alongside telephone interpreting, video remote interpreting (VRI) was considered to be the solution of choice for language-discordant clinical encounters [89], though challenges in building a connection with a patient while interpreting remotely have been documented [87]. Further, VRI requires appropriate technical and spatial arrangements as well as specific skills on the part of healthcare professionals [89] including quality speaker phones and stable phone lines, which are important to avoid technical difficulties. Issues of equity must be addressed ensuring that patients have appropriate access to technology such as phones or computers [2].

Artificial Intelligence (AI) is playing an increasingly important role in facilitating communication between patients who experience language barriers and healthcare providers [90]. Machine translation (MT) tools, where both the patient and the healthcare providers can speak into mobile applications equipped with AI-powered interpreting services, are used to capture and transcribe the speech of each and convert the transcribed text into written and oral language of the other, enabling effective real-time communication. However, research on the use of MT in healthcare is identifying both the specific strengths and limitations of MT, including life-threatening consequences. It is recommended that AI should primarily be used as a supportive tool to facilitate initial communication and provide basic assistance until professional language services can be arranged [91]. Interestingly, a review conducted by Shamsi et al. (2020) found examples where tools such as Google Translate and MediBabble (Tracxn Technologies Limited, San Francisco, CA, USA) may actually improve the quality of healthcare and the level of satisfaction among both medical providers and patients [80]. 

These tools may be beneficial for use by front office staff including for the translation of emails and patient instructions, providing appointments or for understanding written health materials when materials in the patient’s primary language are not available. Where possible, when print materials are provided in multiple languages, they must be written in plain language with consistent messaging from trusted sources as per ISO standards [92]. Once work is carried out to produce them, including translation costs, they should represent common content and ensure quality translation which, when shared widely, can contribute to shared repositories that many providers access, reducing costs and enhancing access. The COVID pandemic demonstrated the need for health systems to be able to quickly provide accurate and updatable public health information accessible by the majority of the public via a variety of platforms. Australian “Health Translations” and Access Alliance “riomix.ca” (accessed on 10 April 2024) are two such examples of how this was achieved. 

#### Comparing Costs/Benefits of Modes of Interpretation

Some studies have attempted to calculate the relative costs and benefits among different modes of professional interpreting, including bilingual healthcare providers, in-person interpreters, telephone interpreting, video interpreting and shared-care networks [52,93].

Fagan et al. estimated the cost of providing telephone interpreters for Spanish-speaking patients in primary care and outpatient department (OPD) hospital settings where there was a high prevalence of such patients and found that two full-time interpreters could be hired for the same cost [94]. Jacobs et al. calculated the average per-minute and per-encounter cost of providing interpreting services via a shared network for various languages [52]. They found that the estimated encounter cost for in-person interpreting services varied widely, depending on whether the interpreter was on staff or under contract. For the former, the estimated average cost was USD 2.65 and, for the latter, USD 15.02. The authors concluded that a shared video and telephone network would enhance the efficient provision and use of these services and reduce health inequities.

Lion et al. conducted a randomized clinical trial to determine the effect of telephone vs. video interpreting on parent comprehension (ability to name the child’s diagnosis), parent-reported quality of communication and interpretation and frequency of lapses in the use of professional interpreters [95]. Parents in the video group were more likely to be able to correctly name their child’s diagnosis (74.6% vs. 59.8% for those in the telephone group) and less likely to report lapses in interpreter use. While mean charges per patient for video interpreting were significantly higher than for telephone interpreting, the authors concluded that video interpreting may be a good investment to improve parent comprehension and lower the risk of communication-related adverse events. 

There is only one Canadian study comparing the costs and benefits of two modes of interpreting. Dowbor et al. conducted a mixed-methods evaluation to compare the impact of Language Service Toronto (LST) [1] to over-the-phone interpretation (OPI) on patients’ and providers’ experiences across a regional health division that included both hospital and community-based health agencies [96]. After the LST program was introduced, the use of face-to-face interpreters decreased from 37% to 24%. Most providers believed OPI services were appropriate for supportive care (90%), followed by acute care (88%), chronic care (86%) and mental healthcare (73%). The program also had a strong positive impact on service processes (e.g., improved patient–provider relationship, increased comfort and privacy levels) and interim outcomes (e.g., increased ability to schedule follow-up appointments and follow healthcare providers’ instructions and increased likelihood to disclose information and ask questions).

As online tools such as Google Translate expand with increased demand related in part to the increased number of newcomers, barriers and costs are projected to decrease and increased research will be required to demonstrate the quality and effectiveness of such services. In any case, such low cost and simple solutions will make it increasingly difficult for providers to justify why they do not make the effort to provide care in non-English languages. Health administrators and clinicians will need to work together to continue to make it the norm that interpreting services are both available and used. 

### 4.4. Providing Quality Care via Medical Interpreting: Intersectional Considerations 

While interpreting services are important for refugees in general and issues such as poor prior physical and mental health, conditions such as internment, post-migration stressors and types of sponsorship may impact this [97,98], it is now widely recognized that intersecting identities (e.g., gender, race, immigration status, abilities, sexual orientation, socio-economic status, etc.) [99,100] can impact health experiences and health outcomes [101,102]. In addition to the language ability of the client, intersectional factors such as gender, culture and other identities need to be considered in the provision of medical interpreting.

#### 4.4.1. Gender

Much research relates to the importance of gender [103,104,105] in medical interpreting. When dealing with certain health areas, such as sexual and reproductive health, it is important to have gender concordance (having the same gender) both between the provider and patient, and/or the interpreter and patient [104,105]. Research has found that women patients in such settings will often “close up” in the presence of male interpreters, which impedes communication [103] and the ability to provide quality care. For refugees who have experienced gender-based trauma, these considerations are particularly acute [104]. In our experience, in many cultures (from parts of the Arabic-speaking world to Old Order Mennonites in Southern Ontario) there are strong norms regarding interactions between women and men, and the extent to which male providers and/or interpreters can interact with women’s bodies. 

#### 4.4.2. Culture

Culture shapes one’s understanding and conceptualization of health. How people speak about childbirth, reproductive health, dementia, death and dying varies among different cultural groups. Medical interpreters have described, for example, how Hmong patients use metaphors and stories to describe symptoms or diseases [106]. Roussos et al. found that cultural concordance may be even more important than gender when trying to establish rapport with patients and enabling them to speak frankly and openly about their health needs and concerns [105].

The importance of culture is increasingly being recognized in medical interpreting practice and the literature. “Cultural awareness” is now included in the *National Standards of Practice for Interpreters in Health Care in the United States* [107]. While interpreters are not required or expected to be of the same culture (and we note that the nebulous concept of “culture” is rarely defined in this work), medical interpreters are expected to make an effort to be culturally aware and to understand the perspectives and worldviews of the patients they are assisting. How this is meant to happen in practice, particularly in the context of very short appointments or emergency situations, is not often discussed. Adding to this complexity is the refusal of some interpreters to participate in discussions about sensitive topics (e.g., medical termination of pregnancy, HIV testing, sexuality or STIs) (pers. comm. authors RT, TH, NA and CB). Therefore, the intersectional identities of both the patient and interpreter must be considered in context, particularly with certain types of care (e.g., reproductive health, mental health, etc.). 

Another complexity in the provision of culturally sensitive interpreting services is the often-incorrect assumption that individuals who speak the same language or who are from the same region are of the same “culture”. We cannot treat people from Hindustani-speaking (Hindi/Urdu) South Asia; Spanish-speaking people from Central America, the Andes, Spain and the Southern Cone of South America; nor Arabic-speaking people from the MENA (Middle East and North African people from Morocco, to Egypt, to Sudan, the Levant and the Arabian peninsula) regions as cultural monoliths, and there may be significant historical class, racial and/or religious intricacies that impact how patients and interpreters interact, even if from the same region speaking the same language [108]. Such intricacies were highlighted by Emma Hadziabdic [67,109,110], who found that patient participants believed that in order to receive quality care and avoid inappropriate treatments, the interpreters should share similar origins and political views, be of the same religion and gender and speak the same dialect. We recognize that in practice, understanding and then meeting these preferences would be prohibitive and may impede access to interpretation. 

#### 4.4.3. Older Adults

Age is another important identity that impacts the provision of interpreting services. Older adults tend to have more complex health needs, including a greater number of chronic conditions that require ongoing monitoring [111] and, consequently, have more regular contact with the healthcare system. The Canadian healthcare system is notoriously complex for older adults to navigate [112], but this is particularly challenging for older refugees and their families [113]. Even though older adults represent a smaller proportion of the refugee population, there is research that demonstrates that this group reports greater challenges navigating the system, experiences greater language barriers and lower health literacy and is dependent on adult children for a range of healthcare needs [114,115,116]. Intersections between age with gender, culture and religion must also be considered. 

Beyond interpreting services, trauma-informed approaches are necessary to prevent harm, but that is beyond the scope of this article.

## 5. Discussion

The provision of medical interpreting services is an essential human right that also makes good health and business sense, benefiting patients, providers and the health system as a whole. Access to interpreting services is even more pronounced for refugee patients experiencing immediate and longer-term health needs. 

However, given the current patchwork of services, this group may be unable to access interpreting services in a consistent, reliable and adequately funded manner. Addressing barriers to the provision of medical interpreting for all Canadians requires government, institutional and provider-level responses. 

### 5.1. Governmental Level

The development of standards is important to protect the health, safety and well-being of all Canadians. Standards ensure consistency, reliability and adequacy of interpreting services by competent, qualified interpreters [117]. Currently, there are no provincial or federal standards regarding the educational qualifications of interpreters and guiding the purchase or utilization of interpreting services. Comprehensive strategies and concomitant funding for interpreting services are needed, ideally at a national or provincial level through the Ministry of Health [118].

A recent petition presented to the Canadian parliament called for federal-level policy to enforce the use of trained interpreters in healthcare settings, an action plan for training and recruiting qualified medical interpreters and mandatory training for healthcare providers on how to effectively work with interpreters [119]. Consistent funding from Immigration, Refugees and Citizenship Canada (IRCC) would assist, and IRCC-IFHP (Interim Federal Health Programme) could consider bulk purchasing, as seen in some provinces. 

Provincial and national professional and regulatory bodies should stress the importance of interpreting services to ensure equitable access to quality healthcare and monitor outcomes to ensure utilization. Practice guidelines such as informed consent and refusal are especially important in healthcare settings such as surgery and emergency medicine where miscommunication and low adherence to treatment may be a matter of life and death. There is also a need to ensure that educational institutions provide training and continuing education programs to ensure that future and practicing health professionals develop a habit and ability to work with interpreters from the outset of their careers.

### 5.2. Organizational Level 

Individual health service providers (HSPs), such as health organizations, are currently involved in the provision of medical interpreting services. ISPs define procurement standards for selecting an ISP, develop the language access plan, collect preferred language data, develop policies and best practices around language support, train staff (from leadership to front-line) on language access and how to work effectively with interpreters, actively offer language support to all clients/patients and dedicate/protect resources for language support initiatives.

The language support strategies and best practices within an organization are part of its language access plan (LAP). Often, as part of its broader health equity activities, an organization’s LAP begins with a needs assessment, starts to take shape by identifying language as a gap or risk and proposes solutions to test or pilot. 

Over the last decade, several health equity initiatives across Canada have identified language as a critical barrier to optimal care [31,118]. 

Supported by feedback from patient and community focus groups, providers and the evidence about the risks and costs of compromised communication, health equity leads have made the case to leadership to endorse and fund pilots to centralize and facilitate access to interpreting services. Common to many of these initiatives have been organizational guidelines about preferred language data collection, *when* to provide interpreting and *who* is qualified to interpret. 

An example of an interpretation implementation guideline was developed by the Waterloo Region Immigration Partnership (January 2023) in Ontario [120]. They established a comprehensive best practice checklist for organizations that included a formal commitment statement; a clear and comprehensive interpretation organizational policy; internal processes in place regarding access to interpreting services; training for staff and volunteers in the awareness of how and when to use interpreting services; a contract with an external professional interpreting organization; communication to patients regarding the availability of interpreting services and data collection. 

The KW4 Ontario Health Team committed member organizations to sign an Interpretation Services Commitment Statement in 2021 acknowledging the importance of interpretation as an obligation of the provider, for reasons of safety, equity and non-discrimination, committing providers to using trained interpretation in “*all medical, legal and other situations that require informed consent, confidentiality, specialized terminology or impartiality*,” in the name of upholding “*our professional, ethical and moral obligations*”. One signatory, St. Mary’s Hospital in Kitchener Ontario, included interpreting services within their 2021 five-year strategic plan and now has this integrated into its electronic medical record (EMR) system. Those facing language barriers are identified on arrival in the emergency department and the requirements for the use of interpretation are flagged for all interactions with the patient, from reception to triage nurse to healthcare provider [121]. 

Pending national and provincial action, organizational action is essential and we cited several of these from hospitals, local cities, community groups and health centres, including what the major barriers are for institutions from financial and logistical challenges. Advocating for support from government and external organizations (e.g., community groups and interpreting service providers) and partnering with communities should occur, especially in settings with high numbers of newcomers. Institutional leaders and administrators need to be educated about costs and benefits in terms of patient outcomes, satisfaction as well as decreased litigation and standards. They may then choose to implement continuous quality improvement initiatives that focus on language access and patient communication with feedback from staff and patients.

### 5.3. Provider Level 

While medical interpreting is a right, an essential element of quality care and can offer cost benefits, it is still underused, both internationally and in Canada across all health sectors [66,122,123,124,125]. For example, publicly funded medical interpreting has been available to family physicians in British Columbia since 2017; however, the uptake of these services remains very low (less than 10% of appropriate cases) [124]. Recently, a retrospective chart review of all patients visiting Kingston Ontario’s Health Science Centre’s emergency department and urgent care between 2012 and 2021 found that less than 1% of patients experiencing language barriers accessed the Language Line interpreting service [126]. These are striking statistics that point to a gross under-use of medical interpreting services in spite of the benefits and rights outlined above. 

Barriers to the use of interpreting services have been identified at several levels. In provinces that cover access to telephone interpreting services, providers may consider the use of these services as onerous or time consuming; not every language group is provided and sometimes services need to be pre-booked. In the case of Quebec, Banques only provide interpretation to French, and VOYCE and R.I.O. Network services only to English. Even in provinces where the cost of interpretation is covered, some providers, especially those in smaller practice settings or with a low proportion of language discordance, may choose to rely on patients’ friends and family, even children, Google Translate or no interpretation whatsoever [96]. On the other hand, most primary care providers working with larger refugee and immigrant populations recognize the critical importance of quality interpreting services.

The reasons that providers report not using medical interpreting services include time limitations, the ease of using family members who are already present, the cumbersome nature of some modes (e.g., delayed telephone and/or internet connections and the need to pre-book in-person interpreters), and in some contexts, the cost of providing interpreting services [123,127]. Though some of these are very real barriers and challenges, they must be overcome. The use of interpreting services takes time, making patient visits longer and potentially more costly up front, but the benefit and potential decreased costs of more effective care are generally seen to outweigh the drawbacks [51,53,56].

Comprehensive training and education to providers regarding the benefits of such interpretation and how to work with medical interpreters will make it easier, less time consuming and more enjoyable. This might include offering workshops, seminars and online courses on cultural competency and effective communication with diverse patient populations and peer-to-peer education. Emphasizing the legal and ethical obligations of healthcare providers to provide language access services to patients experiencing language barriers and reinforcing that working with professional interpreters ensures accurate communication, improved patient safety and outcomes and regulatory and legal compliance. Access is critical and integration in the EMR with regular reminders as to use, administrative support and perhaps performance incentives.

It is important to provide opportunities for patients to be a part of the development and evaluation of interpreting services. Their insights and experiences are invaluable to assist in making policies regarding the implementation of interpreting services. Refugees need to be empowered to advocate for their right to interpreting services [118].

## 6. Recommendations

The provision of medical interpreting for refugees and all Canadian patients experiencing language barriers is not optional, it is essential. The following multi-level recommendations are made.

### 6.1. Governmental Level

The goal of a government health system to be accessible, inclusive and provide high-quality healthcare to all its citizens requires the provision of medical interpreting. At the government level, we recommend that governments do the following: Develop national standards for professional interpreters.Develop comprehensive strategies and concomitant funding models for professional interpreting services to support all healthcare organizations, from health promotion to tertiary care, ideally at a national level, or provincial level through Ministries of Health.Include the mandatory utilization of professional interpreters in Provincial and National Health Care Provider regulatory bodies’ policy and procedure Standards of Practice guidelines.

### 6.2. Organizational Level

A goal of healthcare organizations is to provide a safe, welcoming and culturally responsive environment. At the organizational level, we recommend that organizations do the following:Ensure that their providers are aware of interpreting services and facilitate their access to such services.Establish mandatory training on cultural competence and the importance of interpreting services within the curriculum of all healthcare provider educational programs.Provide written health information in different languages, and different modes (i.e., not exclusively digital).Integrate access to interpreting services in EMR systems with regular reminders as to use.Develop and implement policies and standards of practice related to language access plans.

### 6.3. Provider Level

As providers ourselves, we recognize the many demands on individual providers and clinics. To support and encourage providers in the uptake and use of medical interpreting, we recommend that providers do the following: Receive appropriate compensation for the extra time and effort used in the use of interpreting services.Receive comprehensive training and education regarding the benefits of interpreting services and how to work with medical interpreters and ensure healthcare provider educational institutions and regulatory agencies have national standards to train people on the proper use of interpretation.Receive ongoing seminars and online courses on cultural competency and effective communication with diverse patient populations and peer-to-peer education.Provide opportunities for refugee patients to assist in the development and evaluation of interpreting programs.Empower refugee patients to advocate for their right to interpreting services.Continue to be informed by initiatives such as local healthcare provider champions for refugee health that encourage local colleagues to use these services.Continue to be guided by research addressing how providers make their decisions regarding the use of interpreting services and in what situations such services and, what type of interpreter is used and whether this varies by specialty, while also considering the proportion of patients who do not have access to desired interpreting services, and how often AI, e.g., Google Translate or family and friends are used and why.

## 7. Conclusions

Refugees face individual, institutional and systemic barriers to healthcare including language barriers [128]. The delivery of interpreting services continues to evolve, especially during and following the COVID-19 pandemic, resulting in more funding and support for telephone-based and video-conferencing services [2,129], at least on a temporary basis [87,130]. This has resulted in growing use and encouraging providers to consider (and validate) how and when remote interpreting could be used as an effective alternative to in-person interpreting. However, access varies widely across Canada and is often unavailable to populations, such as refugees who require comprehensive, high-quality medical interpreting services to improve health outcomes. We argue that access to medical interpretation is an essential right founded on four foundational rights: (1) Right to healthcare; (2) Right not to face discrimination; (3) Right to informed consent/refusal and (4) Right not to be harmed. Support for these rights is drawn from UN/WHO documents, professional codes and legal frameworks. Throughout this document, we have summarized the wealth of evidence that medical interpreting improves health outcomes and reduces many of the health disparities faced by refugees. Aside from the ethical and legal arguments in favour of medical interpretation, it is also a financially wise choice for the healthcare system. We have outlined best practices for medical interpreting and provided recommendations for the healthcare system and individual healthcare providers. It is essential that this right be recognized and refugees facing language discordance are provided the access to medical interpreting that they deserve. Neglecting to do so remains a dereliction of medical ethics and represents a moral failing of our society.

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
