# Peer review of "Medical Interpreting Services for Refugees in Canada: Current State of Practice and Considerations in Promoting this Essential Human Right for All"

_ijerph, 2024, doi:10.3390/ijerph21050588_

Round 1
Reviewer 1 Report (Previous Reviewer 3)
Comments and Suggestions for Authors
Even if it is an essay, it must be based on existing literature, but there are too few literature citations. For example, part 1 has only one supporting document [3]. Overall, based on the existing literature, please expand on the story the author wants to tell.
Author Response
"Please see the attachment.

Reviewer 2 Report (Previous Reviewer 1)
Comments and Suggestions for Authors
Great essay, thank you very much for contributing to contextualize the barriers for migrants in Canada
Author Response
Thank you so much for this validation. We do hope it can be published soon.
This manuscript is a resubmission of an earlier submission. The following is a list of the peer review reports and author responses from that submission.
Round 1
Reviewer 1 Report
Comments and Suggestions for Authors
I have conducted a thorough review of your article titled "Medical Interpreting Services for Refugees in Canada: Current State of Practice and Considerations in Promoting this Essential Human Right for All." The paper offers a valuable theoretical perspective on the essential right of interpreters for refugees to access quality healthcare. It is evident that the comments within the paper are well substantiated by relevant references. However, I would like to bring to your attention that the paper lacks a clear methodology and results.
Given the importance and potential impact of the subject matter, it may be beneficial to consider addressing this issue by submitting a letter to the editor. Your insights have the potential to contribute significantly to the field.
Reviewer 2 Report
Comments and Suggestions for Authors
The paper should be submitted as a rapid review. The methods are not explained and the methodology for gathering the evidence is completely lacking. There must be a more structured link between the methods used to gather evidence and the results and evidence synthesis.
However, the topic of this article is relevant and the authors try to address the complexity of this matter. Based on this comprehensive review, the conclusions could offer more focused suggestions for promoting access to medical interpreting services.
Reviewer 3 Report
Comments and Suggestions for Authors
1. This paper is considered to be a literature review paper based on more than 100 documents. However, it is a rather strange paper with no research method. Please create a research method section and describe your research method in detail, including the method of this study, how to search the literature you considered, and key keywords.
2. Clearly state the supporting sources that led to the table on page 3.
3. The table on page 3 contains too many abbreviations, which makes it too unreadable. As much as possible, please refrain from making a table, and explain the abbreviations that could not be used in a limited way in a clear comment.
4. What is the meaning of Yes and No on the table 3?
5. Please present your work throughout the paper to clarify the references. In particular, on page 5, there are a lot of techniques but no sources at all.
Reviewer 4 Report
Comments and Suggestions for Authors
This paper comprehensively deals with the issues surrounding medical interpreting services for refugees in Canada. It sets out the arguments for interpreting in clinical settings as a right for all with language barriers, and concludes with a set of clear recommendations derived from the previous arguments and analysis. However, I believe there is a need for the paper to be improved in two possible areas:
1. The title and introduction focus on refugees, but much of the following text is more general. This is understandable, as the issues around interpreting hold for all community members with language barriers. It would assist the reader and make the argument more consistent throughout if the focus was brought back to refugees at various points in the paper – e.g., what evidence is there that refugees in particular suffer from a lack of interpreting services? In which parts of Canada are refugee interpreting services most needed, in relation to the authors comprehensive coverage of where and how services are delivered? A focus on refugees could also be included in the material on pp14-15. On p2 l72 the authors state ‘we are focussing on the heightened needs of refugees’ but this focus is obscured for much of the paper.
2. The level of detail provided is at times overwhelming. It would make the paper more relevant and engaging for the non-Canadian reader if some of this detail could be summarised (and possibly included in Supplementary Material). The text on pages 2 – 6 is covered in the Table on pp3-4 and could be briefly summarised, making the important point that services are not standardised across the country. There are other parts of the text where the Canadian situation could be summarised and more general points drawn out – e.g., p12 on certification where the acronyms at line 486 are a good example of a place where the reader will struggle.
Overall the paper is well written and clear. There are a few specific points that need attention:
· P2 l73 “in this work ‘ – it is unclear what work is being referred to here.
· P6 l174ff –‘most primary care providers……’ evidence for this statement?
· P6 l188 and p7 l239 – remove the fullstop at the beginning
· P11 l444 ‘We generally recommend……’ who is ‘we’ here and what authority is being invoked here?
· P13 l529 – sense not clear
· P17 l764 - u47p – error.
Comments on the Quality of English LanguagePaper is well written overall. A small number of minor issues are identified in my review.